# The Fate of Antibiotic Impregnated Cement Space in Treatment for Forefoot Osteomyelitis

**DOI:** 10.3390/jcm11071976

**Published:** 2022-04-01

**Authors:** Inha Woo, Jeongjin Park, Hyungyu Seok, Tae-gon Kim, Jun Sung Moon, Seung Min Chung, Chul Hyun Park

**Affiliations:** 1Department of Orthopedics, Yeungnam University Hospital, Daegu 38541, Korea; buonggiorno39@gmail.com (I.W.); wjdwls3912@naver.com (J.P.); rkaldhkthfl4@naver.com (H.S.); 2Department of Plastic Surgery, College of Medicine, Yeungnam University, Daegu 38541, Korea; kimtg0919@daum.net; 3Department of Internal Medicine, Division of Endocrinology and Metabolism, College of Medicine, Yeungnam University, Daegu 38541, Korea; mjs@yu.ac.kr (J.S.M.); smchung@ynu.ac.kr (S.M.C.); 4Department of Orthopaedic Surgery, College of Medicine, Yeungnam University, Daegu 38541, Korea

**Keywords:** forefoot osteomyelitis, antibiotics impregnated cement spacer, diabetic foot, single center study

## Abstract

Forefoot osteomyelitis can be an extremely challenging problem in orthopedic surgery. Unlike conventional methods, such as amputations, antibiotic impregnated cement space (ACS) was recently introduced and perceived as a substitute for amputation. The purpose of this study was to compare clinical features between diabetic and non-diabetic groups and to evaluate the efficacy of ACS in the treatment of forefoot osteomyelitis, by identifying the clinical characteristics of ACS. We inserted ACS into the forefoot osteomyelitis patients and regularly checked up on them, then analyzed the clinical features of the patients and failure reasons, if ACS had to be removed. Average survival rate of ACS was 60% (21 out of 35 cases) and main failure reason was recurrence of infection. There was no significant clinical difference between diabetic and non-diabetic groups. We concluded that ACS could be a possible way of avoiding amputation if infection is under control. ACS seems to be an innovative method with promising results for foot osteomyelitis, but widely accepted indications need to be agreed upon.

## 1. Introduction

Foot osteomyelitis can be extremely debilitating to patients and chronic osteomyelitis remains one of the most challenging problems in orthopedic surgery. Foot osteomyelitis can be caused by various etiologies, and diabetic foot infection is one of the most frequently encountered in clinical scenarios. About 25% of patients with diabetic foot infection will see it spread from the skin to deeper subcutaneous tissues and/or bones, such as osteomyelitis [1,2].

In patients with diabetic foot, osteomyelitis occurs in 10–15% of moderate infections and 50% of severe infections [3]. In particular, the forefoot is the most common site for diabetic foot osteomyelitis and has a better prognosis than the midfoot and hindfoot [4]. Individuals with diabetes have a 30-fold higher lifetime risk of undergoing a lower-extremity amputation compared to those without diabetes [5]. However, so far, many treatment options have been introduced but no significant differences in outcome were associated with any particular treatment strategy [6]. In the case of non-diabetic foot osteomyelitis, a lot of data are required, but it was found that active debridement and dead space control are required, as in diabetic foot disease [7].

The standard of diabetic foot osteomyelitis usually consists of extensive debridement of the infected soft tissues and foci [8]. However, recently, the surgical treatment of foot osteomyelitis has been tried based on conservative surgical treatment [9], in which only the infected bone is removed and the external structures remain. It can reduce loss of the body structures and has cosmetic advantages by avoiding amputation [4,10]. Still, structural instability can be induced due to bone removal and there is a risk of recurrence of infection due to dead space.

To address these complications, antibiotic impregnated cement space (ACS) was recently introduced [11,12,13]. Infected soft tissue and bone are removed and ACS is inserted into the dead space. It serves as a local antibiotic delivery and void filling, and has the advantage of reducing the empty gap of structural deformations, such as shortening of the toes. However, to the best of our knowledge, there are few studies on the results of treatment with ACS for foot osteomyelitis and its long-term prognosis.

We hypothesized that the use of ACS would be an effective method to treat deep infection, reduce amputation rate, and be a permanent substitute in foot osteomyelitis with or without diabetic foot infection. The purpose of this study was to compare the clinical features between diabetic and non-diabetic groups and to evaluate the efficacy of ACS in the treatment of forefoot osteomyelitis by identifying the clinical characteristics of ACS.

## 2. Materials and Methods

### 2.1. Study Subjects

This study was conducted retrospectively with the approval of Institutional Review Board of our hospital, and informed consent was waived because of its retrospective design. From November 2013 to October 2020, a total of 179 patients were gathered who had forefoot osteomyelitis at a single institute. Among these, 42 patients with ACS before were identified. Patients who had received flap surgery or with <6 months of follow-up period were excluded. Finally, 35 cases who consecutively had surgical treatments for forefoot osteomyelitis including metatarsals and phalanges by a single orthopedic surgeon were analyzed. The inclusion criteria were as follows: (1) Patients with forefoot osteomyelitis underwent surgical debridement and ACS implantation. (2) Patients with follow-up more than 6 months. The exclusion criteria included: (1) Patients who received surgical treatments except surgical debridement and ACS implantation. (2) Patients in whom limb salvage was not possible due to severe infection such as necrotizing fasciitis. (3) Patients who needed flap surgery due to insufficient soft tissue coverage after surgery. (4) Patients with follow-up less than 6 months. Finally, 34 patients (35 cases) met the criteria and were included in the study. Among all, 26 were male and 8 were female. The mean age was 61.03 years (34 to 81), and the mean follow-up period was 39.6 months (12 to 100). Figure 1 demonstrates how we selected our study subjects. Of the patients, 27 were diagnosed with diabetes and 8 were not.

### 2.2. Preoperative Evaluations

Chronic osteomyelitis is typically defined by characteristic histopathological findings such as the persistence of microorganisms, low-grade inflammation, the presence of devitalized bone (sequestrum), new bone (involucrum) formed in response to the sequestrum, fistulous tracts (cloacae), and soft tissue involvement [14]. All the patients included in our study were subsequently pathologically diagnosed with chronic forefoot osteomyelitis. Cierny and Mader developed a detailed classification with chronic osteomyelitis appliable to long and large bones. This classification was a combination of four anatomic types (1 to 4) with three physiologic classes (A to C) to figure out the clinical stages and incorporate the prognostic factors [15,16]. All of the patients were associated with type 4, diffusely involving the entire circumference of a segment of the bone. Twenty-seven patients who had diabetes were classified as class Bs and the others were classified as class A.

The evaluation of the ischemic state of the foot was performed in cooperation with the vascular surgery department of this institution. Vascular evaluations were assessed using ankle-brachial index (ABI) and doppler arterial pressure and pulse wave measurement. If necessary, computed tomography angiography was performed. If there were patients who required vascular procedure, then they received proper procedures preoperatively. In our study, none of the patients received vascular interventions except one.

The evaluation of osteomyelitis was performed through plain radiograph, three-phase technetium-99 m bone scan (3-phase BS), and magnetic resonance imaging (MRI). On plain radiograph, osteomyelitic signs including periosteal reaction, sequestration, loss of trabecular pattern, or cortical destruction were observed. In case of diabetic foot osteomyelitis, the most reliable and accurate radiographic sign is the cortical disruption [17]. On 3-phase BS, osteomyelitis was suspected when increasing uptake over all three phases was observed throughout the course of the study. On MRI, the diagnosis of osteomyelitis was made based on altered bone marrow signal and signs of edema and inflammation in adjacent soft tissues. Osteomyelitis was preoperatively diagnosed by considering findings of plain radiograph, 3-phase BS, and MRI.

### 2.3. Surgical Technique

The surgery was performed using spinal anesthesia or nerve blocks. In order to check the soft tissue bleeding, the surgery was performed without the tourniquet if possible. After making proper incision, all infected tissues including skin, tendon, joint capsule, fascia, fibrous tissues, and bone were removed thoroughly until the bleeding tissues were exposed. The resection extent of osteomyelitic bone was determined by considering 3-phase BS and MRI performed before surgery. After resection of the suspected osteomyelitic bone, an extra 2 mm of healthy bone was additionally excised for the prophylaxis of residual pathogens. If possible, the bases of metatarsal and phalangeal bones to which tendons attach were preserved. Resected bone and soft tissues were used for culture and histopathologic examination to confirm the diagnosis of infected pathogen and osteomyelitis.

For the antibiotic mixed cement filling, polymethyl methacrylate premixed with gentamycin was used (Palacos R + G, Heraeus Medical GmbH, Wehrheim, Germany). The antibiotic powder was mixed (typically vancomycin, 1.0 g per package) before combining with the monomer. When cement was semi-rigid but still plastic, it was inserted into the space created after the debridement. Pressure was applied circumferentially from all directions to complete the dead space while maintaining as much anatomical length and support as possible. A free elevator (freer) was used to check that the cement was properly positioned in all sides of the dead space. In addition, physiological saline irrigation was performed to reduce the heat generated when the cement became hardened. If the stability between the cement filling and the bone was judged to be poor, a Kirshner-wire (K-wire) fixation was considered. The skin was sutured with low tension.

If the wound was in good condition, the suture was removed 2 weeks after surgery. In case of recurrence of foot ulcer, acute pain or gross inflammation (redness or swelling around the wound), it was considered a reinfection and additional tests were performed.

The K-wire was removed one month after surgery. The patient was prohibited from weight bearing until the wound healed and movement was restricted by splint immobilization. After all wounds were healed, joint movement and weight bearing were allowed. Shoes or insoles for patients were used as needed. In the case of weight bearing, while checking the patients’ condition, it was adjusted so that partial weight bearing could be fully weight bearing.

### 2.4. Outcome Assessments

Patients were followed up daily during the hospital stay, and after discharge, they visited weekly until the wound was healed or the sutures were removed. After all wounds were completely healed, follow-up was performed once a month for 3 months, followed by follow-up every 3 months thereafter. After discharge, radiographs and gross photos were taken at each visit. Resolution of osteomyelitis was determined by radiographic, clinical, and laboratory changes. “Failure” was defined as progression of osteolytic findings on radiographs, increased inflammatory markers (C-reactive protein, erythrocyte sedimentation rate), or worsening of the wound. Changes in the wound were judged as failure if pus formed, color changed, swelling with tenderness occurred, and this was performed by single orthopedic surgeon.

Initially we first categorized our study groups into two, whether they had diabetes or not. Then we evaluated demographic traits, comorbidities, location of osteomyelitis and longevity of ACS in each group. We also gathered microbiological findings which were obtained during operation. Antibiotics were modified according to these culture results. Subsequently, we compared ACS retention group and ACS removed group. We tried to find differences between two main groups and found failure reasons. It was assumed that the antibiotics used in the ACS failure group would have used so-called “strong” antibiotics, and the injections used in them were also investigated.

### 2.5. Statistical Analysis

For the continuous variables, where the assumption of normality did not appear to be satisfied, Mann–Whitney U test was used. For the categorical variables, Fisher’s exact test was conducted. For all tests, a *p*-value of <0.05, was considered significant. All statistical analyses were performed using IBM SPSS Statistics ver. 21 (SPSS Inc., Chicago, IL, USA).

## 3. Results

We classified the patients as diabetic and then analyzed the demographic features of each. Table 1 shows that no statically significant differences were identified. No association was found between maintenance of final ACS and diabetes and we also acquired microbiological findings (shown in Table 2), which showed that *Staphylococcus aureus* was the most common organism, accounting for 22.8% of total cases. MRSA was also an easily found organism (4 out of 35 cases, 11.4%). All the cultures were obtained during surgery by curettage and biopsy [18] and antibiotics were modified according to an antibiotics susceptibility test.

At the final follow-up, ACS remained in 21 of 35 cases, resulting in a survival rate of 60%. We had 14 failures, defined by the need to amputate the involved part or remove the inserted ACS. In the failure group, the average ACS retention period was 348 days. However, in two cases, the retention period was elongated extraordinarily; therefore, except these two cases, the average retention period was 122 days. Recurrence of infection was the most common cause of ACS removal in eight cases, ACS penetrated the skin in three cases, and soft tissue was not covered with the ACS insertion site in one case. In these cases, the ACS was removed and all surrounding soft tissues were debrided. No additional cement spacer was inserted in the dead space, and no additional bone resection was performed except one case. In this case, he had uncontrolled diabetes and infection was not controlled continuously after removal of cement, so first ray amputation was performed. Previous operations performed before ACS insertion were 2.24 and 1.29, retrospectively, in the ACS retention group and removal group. The ACS retention group showed statically low TBI compared to the other group. Many operations, which implied thorough infection control, were required to obtain a competent ACS result (Figure 2).

These results are shown in Table 3 and Table 4. The location of the lesion was divided into three areas: phalanges, metatarsals and metatarsophalangeal joints involved area. Phalanges were the most common sites where osteomyelitis occurred. The more the metatarsal bone was invaded, the higher the failure probability was. Finally, the list of antibiotics prescribed in the failure group is listed in Table 5. Of these, cephalosporins were the most commonly used.

## 4. Discussion

In our retrospective study, some clinical features were consistent with prior studies. Generally, it is known that *Staphylococcus aureus* is the most common organism of these infections, according to previous studies [19,20]. ACS finally showed about a 60% survival rate, being used as a permanent spacer, which is similar to what Elmarsafi et al. showed. They retrospectively assessed longevity of ACS and two-thirds (20 out of 30 patients) of all patients had successful spacer, with a mean follow-up period of 52 months [21,22]. This prevents the shortening, deformation, dead space, and instability of structures that may occur after debridement surgery, thereby reducing the risk of dysfunction and recurrence of infection [23]. In another retrospective study, Melamed and Peled [24] performed debridement and ACS implantation in diabetic foot patients with osteomyelitis. Out of 23 cases, except for 2 cases, where toe amputation was additionally performed, successful treatment results were obtained without amputation. Park et al. tried to avoid amputation as much as possible, by removing the antibiotic cement between 6 and 8 weeks after surgery and performing an autologous iliac bone graft in the empty space. It was different from our study, in which ACS was inserted semi-permanently as much as possible [12]. There is still a paucity of confirmed data, but recent studies are being published, showing how concomitant antibiotic administration (intravenously or orally, and local antibiotic delivery system) shows good results and reduces the rate of amputation in patients who have osteomyelitis.

To the best of our knowledge, our study, unlike previous studies, is practically the only one that has tried to focus on determining the correlation between ACS survival rate and the presence of diabetes. However, there was no significant difference in the survival period of ACS between the two groups. This implies all forefoot osteomyelitis should be treated, regardless of having diabetes or not.

In our study, we performed limb salvage operations on 34 (35 cases) foot patients with osteomyelitis who required amputation and performed debridement and ACS implantation. All cases showed successful results without amputation, except one case. This means that in forefoot osteomyelitis patients, ACS could be a good possible substitute of removed osteomyelitis sites, without performing additional amputation. This suggests that good results can be expected when ACS acts as a permanent spacer. The most common reason for ACS removal was recurrence of infection. Therefore, to increase the success rate of ACS, the first priority is to prevent recurrence of infection thoroughly.

Effective systemic antibiotic therapy is still an essential component of the most curative treatment regimens for osteomyelitis [25]. Accompanied by the traditional intravenous route of giving antibiotics, ACS acts as a local antibiotic delivery system, which serves higher local antibiotics concentration, longer-acting duration and fewer systemic side effects. ACS also works as a bone substitute, which fills dead space.

There are still limitations that remain unanswered. As this was a retrospective study, a control group was not included. Furthermore, ACS seems to be an innovative method with promising results for foot osteomyelitis, but widely accepted indications need to be agreed upon. After the antibiotic is released from the antibiotic-mixed cement, the cement can be used as a harbor, especially when wound healing is incomplete. There may be a risk of secondary infection by resistant strains. Wong and Hui pointed out the safety of the semi-permanent preservation of ACS was not guaranteed [5,26]. Generally, patients who suffered from osteomyelitis tend to have unexpected progress during treatment. In these cases, long-term follow-ups were required if the infection recurred, which made it difficult to set the definite indications.

The difference in the number of patients between the diabetic and non-diabetic groups can also be viewed as another limitation. It was hypothesized that the outcome would be worse in the presence of diabetes, and we conducted additional studies to confirm this. However, since the sample size is small, this may be biased. The indications for performing ACS in osteomyelitis were not large. Therefore, it was difficult to conduct a study targeting a large number of patients. Therefore, the next step should be a multicenter study or random controlled test (RCT) to provide answers [27].

## 5. Conclusions

In conclusion, our findings show ACS could be a good treatment option, serving as a possibly permanent bone substitute, giving stability, but also serving eluting antibiotics locally. Our study has shown that ACS in the foot is both safe and durable. We acquired successful treatment results with ACS, which finally avoids amputation. However, there are still limitations, such as the proper ACS indication or how much soft tissue should be secured. With all these efforts, we expect that ACS after bone resection is a viable limb salvage tool, as demonstrated by many noteworthy study results.

## Figures and Tables

**Figure 1 jcm-11-01976-f001:**
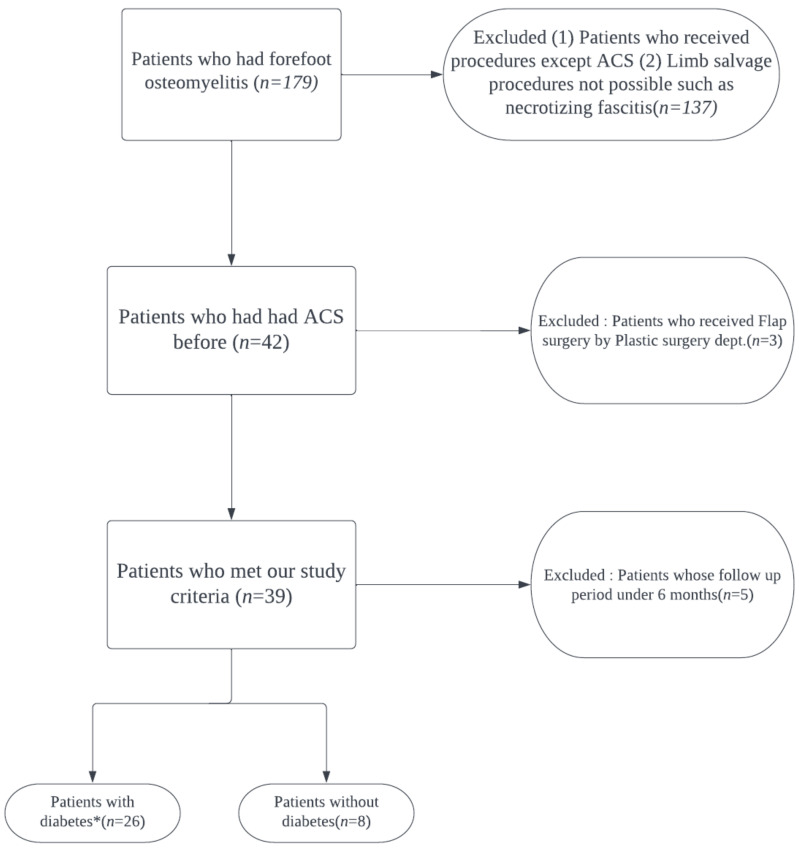
Patient selection flow chart. (* One patient belonging to the diabetic group occurred on the left and right for different periods, respectively, and was considered as each case).

**Figure 2 jcm-11-01976-f002:**
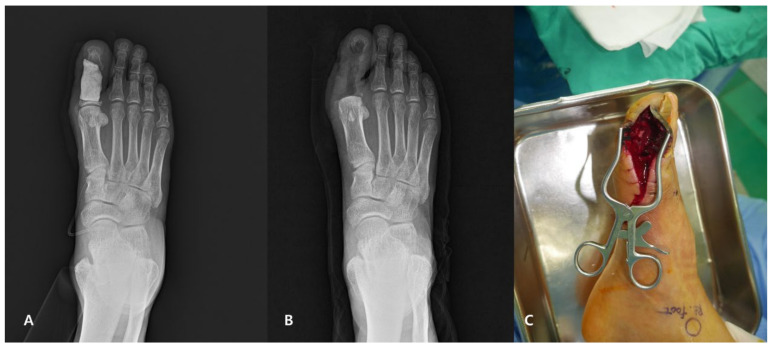
A 62-year-old male patient with diabetes who had had first phalanges infection and ACS insertion and received additional surgery due to uncontrolled infection. (**A**) The X-ray presentation of ACS insertion initially. (**B**) The radiograph after first ray amputation performed. (**C**) Pus filled with dead space after ACS was removed intraoperatively.

**Table 1 jcm-11-01976-t001:** Demographic features and final status of ACS between diabetic and non-diabetic group of the patients.

Demographic Data (*n* = 35)			
Variable	Patients with DM (*n* = 27)	Patients without DM (*n* = 8)	*p*-Value
Age (yr)			0.83
Median	64	60	
IQR	60.0–75.3	51–67	
Sex			1
Male	21	6	
Female	6	2	
Follow-up (months)			0.28
Median	16.7	34	
IQR	15.6–39.8	17–48.6	
BMI (kg/m^2^)			0.529
Median	22.7	22.7	
IQR	21.4–24	21.3–27.3	
Comorbidity			
Hypertension	14	2	0.244
Renal disease which requires hemodialysis	6	0	0.299
Cardiac diseases	7	0	0.16
Location			0.313
Phalanx (joint not involved)	15	7	
joint involvement	10	1	
Metatarsals (joint not involved)	2	0	
Maintenance of ACS			0.858
Remained	15	6	
Removed	12	2	

Abbreviatoin: IQR (interquartile range), BMI, body mass index. DM, diabetes mellitus.

**Table 2 jcm-11-01976-t002:** Microbiological results from cultures proceeded at operation rooms.

The Microbiological Findings from Cultures at Operation Room	
Organisms	Number
*Staphylococcus epidermidis*	2
*Staphylococcus aureus*	8
*MRSA*	4
*Enterococcus faecalis*	4
*Enterobacter cleacae*	1
*Pseudomonas aeruginosa*	4
*Proteus mirabilis*	2
*Streptococcus species*	3
*Serratia marcescens*	1
*Klebsiella pneumoniae*	1
*Morganella morganii*	2
no growth	3
multiple organisms	0
Total	35

Abbreviation: MRSA, Methicillin resistant staphylococcus aureus. Multiple organism group is not counted into individual group.

**Table 3 jcm-11-01976-t003:** This table shows comparison between ACS retention group and ACS removed group.

Comparing between ACS Retension Group and ACS Removed Group			
Variable	ACS Retension Group (N = 21)	ACS Removed Group (N = 14)	*p*-Value
Age (yr)	60.00 (52.5–68)	58.00 (53.50–73.25)	0.96
mean Ankle brachial index (ABI)	1.16 (1.08–1.22)	1.096 (1.13–1.17)	0.503
mean Toe brachial index (TBI)	0.77 (0.48–0.89)	1.01 (0.93)	0.03
Numbers of previous surgeries before ACS insertion	0 (0–2.5)	0 (0.5–2)	0.594

Median (IQR). Patients evaluated with 3D angio CT or unable to be evaluated were excluded.

**Table 4 jcm-11-01976-t004:** This table shows clinical features of ACS failure group.

Features of Why and Where ACS Were Failed, Time to Failure (%)	
Variables	
Retained ACS cases	21 (60%)
ACS removed cases	14 (40%)
Additional amputation needed	1 (7.1%)
Reasons why ACS were removed	
infection not controlled	8 (57.1%)
wound problem (such as skin penetraion, not healed op scar)	5 (35.7%)
difficulty in walking	1 (7.1%)
Location of ACS being failed (total number of each group)	
phalanx	6 (out of 22, 27%)
MPJ (joint involved)	5 (out of 11, 45%)
metatarsals	2 (out of 2, 100%)
Other *	1 (out of 1, 100%)
Time to failure in ACS removed groups (days)	
mean	348
Range	2 to 2332

Abbreviations: MPJ Metatarsophalangeal joint. *: This patient had osteomyelitis on 1st metatarsal head and 5th distal phalanx.

**Table 5 jcm-11-01976-t005:** Antibiotics list of systemic antibiotic therapy.

Category of Antibiotics Prescribed to the Failure Group	
Variable	Number
Cephalosporins (first to third generation)	5
Piperacillin/Tazobactam	1
Floroquinolones	4
Vancomycin	1
Tigecycline	1
Carbapenem	2
Total	14

## Data Availability

Not applicable.

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
