# Peer review of "The Fate of Antibiotic Impregnated Cement Space in Treatment for Forefoot Osteomyelitis"

_jcm, 2022, doi:10.3390/jcm11071976_

Round 1
Reviewer 1 Report
Interesting work but needs a revision to better specify some missing data:1) Specify the outcomes between diabetic and non-diabetic group
2) Define type of osteomyelitis, if acute or chronic
3) The localization of osteomyelitis which involves in 22/35 cases the phalanges while involves the metatarsals only in 2 cases (only small segments ??) 4) Define wound healing and back to walk times
5) Define which systemic antibiotic therapy was adopted, above all in cases of removal of the ACS due to recurrence of infection
6) Define if the only patient undergoing additional amputative surgery was in the group of diabetic or non-diabetic patients
7) Specify if instrumental examinations were performed in the follow-up (to check the state of the ACS, X-ray?)
8) Specify the treatment and how the dead space was filled after the removal of the ACS
9) Specify which site was most subject to the removal of the ACS and in which sites it was left permanently
10) Specify how the diagnosis of osteomyelitis resolution was made (e.g. instrumental? Blood tests?)
Reviewer 2 Report
The fate of antibiotic impregnated cement space in treatment for forefoot osteomyelitis
I consider that this study deals with an interesting topic within the field of diabetic foot osteomylitis. Some major changes could improve this retrospective study.
Introduction:
Line 40 . ….It can reduce loss of the body structures and has cosmetic advantages by avoiding amputation… authors should be consider that conservative surgery also has biomechanic advantages for patients.
I recommend these paper:
Aragón-Sánchez J, Lázaro Martínez JL, Álvaro-Afonso FJ, Molines-Barroso RJ. “Conservative surgery of diabetic forefoot osteomyelitis. How can I operate on this patient without amputation?” Int J Low Extrem Wounds. 2015;14(2): 108-31. 2014;13(1):27-32. Doi. 10.1177/1534734614550686. JCR (2015) Factor de impacto: 1,366. Quartil: Q3, Ranking: 111 de 200
Lázaro Martínez JL, García-Madrid M, García-Álvarez Y, Álvaro-Afonso FJ, Sanz-Corbalán I, García Morales E. Conservative surgery for chronic diabetic foot osteomyelitis: Procedures and recommendations. Journal of Clinical Orthopaedics and Trauma (2020), doi: https://doi.org/10.1016/j.jcot.2020.12.014.
Material and methods
Line 76 Patient selection flow chart: at the end of the flow chart should be consider separate patient with and without diabetes
Line 78… for Cierny and mader……… For Futures studies I recommend revise this work:
Histopathologic Characteristics of Bone Infection Complicating Foot Ulcers in Diabetic Patients. Matilla et al. 2013
Line 91-99: Some references should be included regarding the evaluation of osteomyelitis… For instance:
Álvaro-Afonso FJ, Lázaro-Martínez JL, García-Morales E, García-Álvarez Y, Sanz-Corbalán I, Molines-Barroso RJ. Cortical disruption is the most reliable and accurate plain radiographic sign in the diagnosis of diabetic foot osteomyelitis. Diabet Med. 2019 Feb;36(2):258-259. doi: 10.1111/dme.13824.
I consider that some figures regarding the surgical techinque, the antibiotic impregnated cement or the splint immobilization of patients could improve the methods section.
Authors should please provide a sample size calculation in the methods section. This is important to show, whether the power was enough to answer the aim of the study.
Result
Line 144. Misprint “T” We categorized….
Line 145. No variables were found that 145 showed a significant difference between the two groups…. Authors should be consider that the main diference among groups is the sample size (27 versus 8). For this situation is important the sample size calculation.
Line 148.
…These results showed that they were consistent with the data of several previous studies[15, 16]… this is discussion. In results section authors should show results without comparison with other studies.
Line 150-151. All the cultures were obtained during surgery by surgical swab…authors should be consider that this is a limitation. Swab is not recomended to obtain culture in this type of infection. Revise:
IWGDF Guideline on the diagnosis and treatment of foot infection in persons with diabetes: obtain a sample for culture by aseptically collecting a tissue specimen (by curettage or biopsy)
A table with the antibiotic precribed in the study population could improve the results
Line 157-168. In consider that authors should explain in methods section: retentionn group, removal group, failure group. On the other hand The authors have previously segmented the population into diabetics and non-diabetics; however, nothing is mentioned about this variable in the results related to surgeries.
Discussion
Line 180-186. Is introduction section… The discussion should start by mentioning the main result of this retrospective study and then it should be compared with previous studies.
Line 202.204 …And from our study, ACS retention period in diabetes and non-diabetes group showed no significant difference, which implies all forefoot osteomyelitis should be treated regardless of having diabetes or not… I think that these results are missing in result section
Line 205 Misprint: opertion
Line 215-218 It is important that the authors take into account the systemic antibiotics prescribed in the study population
In general, I think the discussion is short and should focus on the main results of the study.
Round 2
Reviewer 1 Report
I don't have other comments. The authors' revision is sufficent.
Author Response
Dear Editors
Thank you for taking your time to review our manuscript.
Sincerely,
The Authors
Reviewer 2 Report
I consider that authors have considerably improved the paper.
Revise flow chart
27 patients with diabetes + 8 patients without diabetes= 35 patients.
Nevertheless patients who meet your study criteria are 34. (patients or cases/foot)
Author Response
Dear Editors
Thank you for taking your time to review our manuscript. We deeply appreciate all of your thoughtful responses. We have revised manuscript according to your valuable comments. Detailed responses to the reviewer’s comments are submitted below.
I revised the flowchart as you pointed out.
one of the patients in diabetic group occured both left and right for differenet period retrospectively so that it was counted as different cases.
Sincerely,
The Authors
